# Dynamic Thermomechanical Analysis on Water Tree Resistance of Crosslinked Polyethylene

**DOI:** 10.3390/ma12050746

**Published:** 2019-03-05

**Authors:** Kun Sun, Junqi Chen, Hong Zhao, Weifeng Sun, Yinsheng Chen, Zhongming Luo

**Affiliations:** 1The Higher Educational Key Laboratory for Measuring and Control Technology and Instrumentations of Heilongjiang Province, Harbin University of Science and Technology, Harbin 150080, China; sunkun1982@126.com (K.S.); chen_yinsheng@126.com (Y.C.); luozhongming@hrbust.edu.cn (Z.L.); 2National Experimental Teaching Demonstration Center for Measurement and Control Technology and Instrumentation, Harbin University of Science and Technology, Harbin 150080, China; 3Key Laboratory of Engineering Dielectric and its Application, Ministry of Education, College of Electrical and Electronics Engineering, Harbin University of Science and Technology, Harbin 150080, China; hongzhao@hrbust.edu.cn (H.Z.); sunweifeng@hrbust.edu.cn (W.S.)

**Keywords:** polyethylene, UV irradiation crosslinking, water tree resistance, dynamic thermomechanical analysis

## Abstract

The water tree resistance of crosslinked polyethylene (XLPE) initiated by ultraviolet (UV) irradiation technique is investigated through a water blade electrode method, and the effects of the mechanism of UV irradiation crosslinking on inhibiting water tree growth are revealed with dynamic thermomechanical analysis (DMA). The accelerated water tree aging experiment shows that UV irradiation crosslinking inhibits the growth rate of water trees, and the water tree length and width is reduced with the increase of the crosslinking degree of XLPE. The DMA result demonstrates that the molecular activity of the amorphous phase in XLPE as represented by polyethylene β-relaxation is gradually intensified with the increase of the crosslinking reaction. Combined with the fatigue mechanism of water tree growth in semi-crystalline polymers, it is suggested that the UV irradiation crosslinking reaction can significantly improve the anti-water-tree performance of linear low-density polyethylene (LLDPE). The crosslinking bond in the amorphous phase of UV-photoinitiated crosslinking polyethylene can produce a large number of cross-connected polymer chains, by which the length of fiber is obviously increased, leading to an reduced force from the micro-water beads onto the crack tip and thus decreasing the rate of the material being destroyed by micro-water beads.

## 1. Introduction

Water tree is a degradation phenomenon for insulation material. It consists of large numbers of voids filled with water inside. It is generally believed that the generation and development of water trees in insulation material is the result of long-term interactions by a nonuniform electric field and infiltrated water. Crosslinked polyethylene (XLPE) is the primarily used insulation material for power cables due to its excellent electrical and mechanical characteristics. Although the lifetime of XLPE power cables is generally 20–30 years in theory, the poor operation conditions will lead to the accelerated aging of the cable insulation such that the actual service lifetime of the cable is much lower than the theoretical prediction. The operation experience shows that the water tree is the potential threat to the XLPE insulation in the initial deteriorating process and the main cause of insulation aging of XLPE power cables, especially for the medium voltage power cables [1,2,3]. Understanding the water tree growth mechanism in insulating XLPE and improving its water tree resistance for the long-term safe operation of power cables is urgently needed.

After the crosslinking reaction, the anti-water-tree resistance of polyethylene will be remarkably improved by producing a large number of tie molecule chains in the amorphous phase due to the existence of crosslinking bonds. Therefore, the ductility of the amorphous phase in XLPE is considerably increased so as to impede the slippage of the lamellar crystal caused by the action of micro-water beads, reducing the growth of the water tree [4,5]. Nevertheless, these research achievements need to be further analyzed. It is still urgent to study how the linking molecular chains formed in the amorphous phase can alleviate the impact force of micro-water beads on the materials, to further understand the mechanism of water tree growth in semi-crystalline polymers.

Dynamic thermomechanical analysis (DMA) is an efficient method to obtain the viscoelastic properties of polymer-based materials, which can effectively reveal the changing law and causal mechanisms of material mechanical properties, and present the detailed relationship between molecular motion and macroscopic properties of materials on a deeper level [6]. Because the resistance of polyethylene to the water tree is closely related to its molecular movement and aggregation structure [7,8,9,10], DMA can be employed to characterize the microstructure of polyethylene before and after crosslinking reaction and explain the intrinsic mechanism of crosslinking reaction to improve the water tree resistance of polyethylene. The motive of this paper is to investigate above mentioned issues by exploiting DMA method.

## 2. Experiments and Characterization Method

### 2.1. Sample Preparation

The mechanism of UV irradiation crosslinking polyethylene reaction mainly includes the initiation, propagation, and termination process. The reaction process is as follows [11]:

Initiation:(1)A→(A)*1→ISC(A)*3→deactivationA
(2)(A)*3+P→A•+P•
(3)(A)*3+T→A•+T• Propagation:(4)$$T+P^{•}\to P-T^{•}$$
(5)$$T+T^{•}\to T-T^{•}$$ Termination:(6)$$P^{•}+T^{•}\to P-T$$
(7)$$P^{•}+P^{•}\to P-P$$
(8)$$P^{•}+P^{•}\to P=(-H)+P(+H)$$
(9)$$P^{•}+A^{•}\to P-A$$ where *A*, *P* and *T* denote the photoinitiator, polyethylene, and crosslinking agent, respectively. First, the photoinitiator absorbs the UV photons to be excited into the triplet state through intersystem crossing (ISC) and captures both hydrogen atoms of polyethylene and the crosslinking agent. The produced radical of the crosslinking agent then expedites the formation of free radicals in the polyethylene so as to accelerate the crosslinking reaction. The existence of the crosslinking agent substantially increases the rate of crosslinking reaction as indicated in Equation (4). Finally, the polyethylene radicals are linked together to generate a crosslinking polyethylene. Therefore, the higher the content of photoinitiator participating in the crosslinking reaction, the greater the crosslinking degree in XLPE.

The XLPE samples were prepared using a UV irradiation crosslinking technique (UV-photoinitiated XLPE) with pristine materials being composed of linear low density polyethylene (LLDPE) of model 7042 as the base resin, benzophenone (BP) as the photoinitiator, and triallyl isocyanurate (TAIC) as the crosslinking agent. First, LLDPE, BP, and TAIC were melted and blended using a torque rheometer (Harbin Hapro Electric Technology Co., Ltd., Harbin, China) to obtain the crosslinkable materials. The rotation mode of screws in the torque rheometer was counter-rotating with the rotational speed of 60 rpm at the mixing temperature of 160 °C for 20 min in order to ensure BP and TAIC was dispersed uniformly in polyethylene. The components of the melting mixtures for fabricating XLPE with various crosslinking degrees consisted of 1 wt % TAIC and individual 1, 2, 3 wt % BP. Further, the mixed materials were compressed into the molded samples with the shape and size as needed for the subsequent experiment using a plate vulcanizing machine under 160 °C and 15 MPa. Then, an array of UV LED elements with a 365 nm wavelength and 100 W electric power was used as the UV light source to irradiate crosslinking the compressed samples, which were laid on a heating pedestal of 170 °C to keep them in a molten transparent state. Eventually, the fully crosslinked samples after accomplishing UV irradiation crosslinking were placed into a vacuum oven at 80 °C for three days to completely remove the residual stress and by-products. The obtained XLPE materials are represented by XLPE-0 wt % BP (pure LLDPE), XLPE-1 wt % BP, XLPE-2 wt % BP, and XLPE-3 wt % BP, respectively. 

### 2.2. Crosslinking Degree Characterization

The crosslinking degree of the obtained materials were characterized by a heat elongation test (refer to IEC 60811-507: 2012 [12]) and gel extraction method (refer to ASTM D 2765-2011 [13]). The detailed test procedure of heat elongation test is as follows: First, the sample was compressed into a “dumbbell” shape with a thickness of 1 mm and a gauge length of 20 mm, and a load with 81.6 g was hung at the bottom of the sample. Further, the loaded samples were placed in an oven at 200 °C for 15 min. Finally, the length of the sample was recorded and its elongation was calculated according to Equation (10):(10)$$\eta =\frac{\left(L^{\prime }-L_{0}\right)}{L_{0}}\times 100\%$$ where η denotes elongation, *L*′ denotes the length of sample after heating, and *L*_0_ denotes the gauge length.

The detailed test procedure of gel extraction method is as follows: First, a sample of approximately 0.3 g was placed into a holder made of 120-mesh stainless steel cloth. Further, the holder with the sample inside was immersed in xylene solution, which was heated to 145 °C for 12 h in order to fully extract the crosslinked section of sample. Eventually, the extracted sample was placed in a vacuum oven at 150 °C for 12 h to volatilize the residual xylene solution and then calculate the gel content according to Equation (11):(11)$$\varphi =\frac{W^{\prime }}{W}\times 100\%$$ where ϕ is the gel content, and *W* and *W*′ are the weight of sample before and after the gel extraction test.

### 2.3. Accelerated Aging Experiment of the Water Tree

In order to avoid the problem of the conventional water needle electrode method in the cultivation of the polymer water tree, the water blade electrode method was utilized to perform the accelerated-aging test of the water tree [14], as is schematically shown in Figure 1. First, one cut was made by using a blade with a 0.03 mm thickness perpendicular to the surface, leaving a residual thickness between the blade and the other surface of the specimen of 2 mm, and then the vacuum coating system was used to evaporate a film of aluminum electrode on the other surface of sample as the grounding electrode for the applying voltage. Finally, in order to eliminate the residual air in the region of the blade defect, the whole experimental installation was put in a vacuum oven for 30 min before implementing the test. The blade defect was controlled to 40 mm × 0.03 mm in size with a 0.01 mm curvature radius of the cutter tip, and the distance of 2 mm from the tip of the blade defect to the other side of the sample. The high frequency voltage was applied with the amplitude and frequency of 4 kV and 3 kHz, respectively. The electrolyte solution was selected as a NaCl solution with a concentration of 1.8 mol/L. After persisting for 7 days of the accelerated water tree aging test, eight sheets of aged sample of 120-μm thickness each were obtained by slicing along the aged blade defect, which were then placed in a methylene blue solution (5 wt % methylene blue) at the temperature of 90 °C for 4 h to adequately dye the region of the water tree. At last, the aged sheet samples were cleaned up with an ultrasonic cleaning instrument and characterized by observing the morphology and size of the water tree with an optical microscope.

### 2.4. Tensile Property Test

The multifunctional electronic tensile testing machine-CMT6000 (manufactured by MTS Industrial Systems Co., Ltd., Shenzhen, China) is employed to measure the mechanical properties of the samples according to SAC Publication GB/T 1040.2-2006 [15]. The test samples were molded into a “5A” dumbbell shape with 1-mm thickness and 20-mm gauge length. The tensile displacement rate of samples was set to 5 mm/min. The engineering stress–strain curve measured during the elongation process using a tensile force machine cannot sufficiently and accurately explain mechanical properties of tested materials. Therefore, the directly measured engineering curves were transformed into a true stress–strain curve by calculating the real stress and strain with the following formulas:(12)$$\sigma =\sigma^{\prime }\left(1+\varepsilon^{\prime }\right)$$
(13)$$\varepsilon =\ln(1+\varepsilon^{\prime })$$ where σ and σ′, respectively, denote true and engineering stresses, ε and ε′ represent true and engineering strains, respectively [16].

### 2.5. Dynamic Thermomechanical Method

The viscoelastic properties of samples were tested with a DMA-Q800 dynamic thermomechanical analyzer fabricated by TA Instruments Company of Delaware, OH, USA. The test was implemented for the samples in a size of 15 mm × 4 mm × 1 mm under tensile mode with the target amplitude, frequency, static force, and dynamic force being set as 15 μm, 1 Hz, 0.375 and 0.3 N, respectively. The sample was initially cooled down to −50 °C for 5 min, then the storage modulus *E*′, loss modulus *E*″, and loss factor tanδ = *E*″/*E*′ were measured in the temperature range of −50 to 100 °C with a linear heating rate of 3 °C/min.

## 3. Results and Discussion

### 3.1. Crosslinking Degree Results

The elongation and gel content of samples obtained through the heat elongation test and gel extraction method, respectively, are listed in Table 1. With the increase of BP content, the thermal elongation and gel content of UV-light XLPE increased and decreased, respectively, implying that the crosslinking degree was significantly improved with increasing BP content. Thus, it is reasonable to use these samples in the experiments described in the following.

### 3.2. Water Tree Morphology

The higher the content of BP, the greater the crosslinking degree, and the smaller the length and width of the water tree of LLDPE. After the crosslinking reaction initiated by UV irradiation, the water tree resistance of LLDPE was evidently improved with the increment of crosslinking degree by raising BP content, as illustrated in Figure 2 and Figure 3 for the water tree size and morphology in various samples. The average length of the water tree was larger than the width owing to the higher electric field in the perpendicular direction of the blade defect than that in the parallel direction, which was more apparent for a higher crosslinking degree. Meanwhile, the water trees represented a considerable discrepancy in morphology for different BP content. The substantial branch configurations appeared in the water tree of XLPE-0 wt % BP, which attenuated in density with the increasing BP content and even disappeared in the water tree of XLPE-3 wt % BP.

### 3.3. True Stress–Strain Characteristics

The true stress–strain curves of samples obtained from mechanical tensile tests are plotted in Figure 4 and the corresponding elastic modulus, fracture stress, broken elongation, and strain hardening index of each sample are listed in Table 2. The true stress–strain curves were fitted using the Ludwig equation to achieve the strain hardening index [17], with the fitting formula as follows: (14)$$\sigma^{\prime }=a+b{\left(\varepsilon^{\prime }\right)}^{n}$$ where *a*, *b*, and *n* are fitted parameters, and *n* particularly symbolizes the strain hardening index, which can represent the ability of a uniform deformation during a tensile process. The high value of strain hardening index means that it is not easy for the material to reach the dispersion instability. With the increase of crosslinking degree, the elastic modulus and strain hardening index of XLPE monotonously decreased and increased, respectively, while the fracture stress and broken elongation increased first and then decreased.

Since the molecular chains of crystallization arrange in a concentrating order and hardly orientate under external force, less plastic deformation will occur for higher crystallinity [18]. The crosslinking reaction can inhibit the ordered arrangement of macromolecular chains in the crystallizing process, resulting in the abatement of crystallinity and elastic modulus of XLPE. When the crosslinking degree is high, the decrease of breaking stress and breaking elongation occurs, which is commonly referred to as “over-crosslinking” [18,19].

During the crosslinking process, the LLDPE radicals combined with each other via covalent bonds, which was also accompanied by the breaking of chemical bonds in macromolecular chains as the degradation reaction. The XLPE with lower crosslinking degree represented higher tensile strength in which the bonds of macromolecular chains were seldom broken, while the degradation reaction of macromolecular chains occurred significantly when there was an increasing content of crosslinking agent, resulting in a reduced breaking stress and breaking elongation. The crosslinking reaction engendered a large number of interconnecting molecular chains in the amorphous phase of LLDPE and thus increased the interaction between adjacent lamellae, which could impede the lamella slipping caused by the electric stress from microbeads under a high electric field. Therefore, the strain hardening index rose up with increasing BP content, implying the density of amorphous molecular chains was increased via the crosslinking reaction.

### 3.4. Viscoelastic Properties

As seen the results of DMA for materials shown in Figure 5, the loss modulus peak for −20 to 30 °C and the loss factor peak for 70 to 90 °C were, respectively, called β and α relaxation peaks. The storage modulus being similar to the elastic modulus represented the material rigidity and decreased with increasing crosslinking degree as explained in the previous section.

β-relaxation is also known as the high temperature glass transition. It is considered that β-relaxation in semicrystalline polymers originates mainly from micro-Brownian motions of the interconnecting molecular chains and chain rings bounded to lamellae [20]. The value of the loss modulus directly characterized the relaxation activity. As a result of the crosslinking reaction, a large number of linked molecular chains were produced between the polyethylene lamellae, which inevitably encountered more damping in relaxation process. Consequently, the crosslinking degree rising definitely led to a higher loss modulus. Furthermore, the relaxation movement of molecular chains in the amorphous phase bore greater hindrance from the crystallization for higher crystallinity, thus needing a higher temperature (molecular kinetic energy) to fulfill relaxation. Hence, the descending crystallinity caused by raising the crosslinking degree of XLPE presented the β-relaxation peak shifting to a lower temperature.

α-relaxation is a complex multifold process that derives from the rotation, slippage of the lamellar folding chain, and the movement of molecular chains on the surface of the sheet crystal [21]. Similar to β-relaxation, the crystallinity had a great effect on relaxation temperature. The higher crystallinity led to the greater thickness of lamellae and resulted in higher relaxation. Therefore, the UV crosslinking reaction could obtain lower β-relaxation temperatures. In addition, the relaxation motion of macromolecular chains was hindered during the relaxation process, which led to the decrease of relaxation strength of LLDPE after crosslinking.

### 3.5. Analysis and Discussion

Although there is no uniform conclusion on the mechanism of polymer resistance to water-tree aging, the theory of mechanical fatigue has been approved by most researchers [22,23,24]. The theory of mechanical fatigue means that microbeads will gradually destroy the microstructure of materials under the electric field over a long time, and eventually accumulate in the preformed pores so as to engender water-filled micropores. Based on the present theory of mechanical fatigue, the stress cracking and crystallization failure account for the essential explanation of initiation and growth of the water tree.

It has been found that many characteristics of cracks caused by sustained stress in polymeric materials are similar to those of a water tree. Therefore, the mechanism of the growth of a water tree is proposed by means of research results for the crack [25,26]. Figure 6 shows the structural features of a crack in polymeric materials when the sample is stressed by the water droplets, where *F* is the stress exerted on the crack by water droplets, *h* is the major semi axis, and *l* is the radius of curvature at each end of the major axis. The partial areas that coexist the microcavities and fibrils at the tip of cracks are crazes. The length of the longest craze is approximately 2*l*. In the position close to the water area, many microcavities are generated under the action of water droplets and the craze will be enlarged. When the growth of the craze reaches to such an extent that the fibrils at the tip of the craze are fractured, the crack is formed and the more water droplets will migrate to the material so that the water tree will be formed. The stress at the crack tip on the first craze *F*_fibril_ can be described using the following formula:(15)$$F_{\mathrm{fibril}}\sim F\left(1+2\sqrt{\frac{h}{l}}\right)$$

It is noted that once the craze fibril rupture initially appears, the stress in the adjacent fibril will increase due to the increasing of *h* and speeds up the rupture rate of the fibrils. This could explain why the water tree growth will accelerate instead of stop once they appear. Yanagiwara assumed that the stress exerted on the crack by water droplets is extremely close to forces exerted on water by a non-uniform electric field [25], as is accurately obtained by the following equation:(16)$$F=F_{s}+F_{T}+F_{m}$$ where *F*_s_ is the surface tension of water, *F*_T_ is the force of thermal expansion of water due to selective heating, and *F*_m_ is the electrostriction force (Maxwell force). The three forces mentioned above are related to the solution concentration, defect type, temperature, electric field strength, etc. As the four samples have the same initial experiment condition in this paper, it is reasonable to conclude that there is no obvious difference in the three forces mentioned above, along with *h*, among the samples. Therefore, the value of the fibril is only influenced by *l*.

As mentioned above, the fibril length in semi-crystalline polymer is determined by the “fluidity” of the sample being linked to the amorphous phase mobility, where more material can be sucked into the fibrils, which in turn increases *l* when the amorphous phase of semi-crystalline polymer is mobile [26]. As mentioned above, the amorphous phase mobility of semi-crystalline polymer can be monitored through the β relaxation intensity via DMA. The higher the intensity of β relaxation in the material, the better the mobility in the amorphous phase. Figure 7 shows the relationship between β relaxation intensity (including *E*″ and tanδ) and water tree size. The value of the tanδ took the temperature corresponding to its loss modulus. It is obvious to see that the water tree size (consisting of length and width) decreased with the increasing of β relaxation intensity. Therefore, it can be concluded that the UV-photoinitiated crosslinking could enhance the mobility of amorphous phase in LLDPE, leading to the increase of the fibrils’ length. The *F*_fibril_ was lower in the material with a higher crosslinking degree, leading to the slow growth rate of the crack and inhibited the development of water tree.

Although the relationship between fibril length and water-tree resistance have been observed extensively when examining different cross-linking degree of XLPE samples, it is still unclear why the tie molecule can increase the fibril length. However, this point can be elucidated from the molecular topology of semi-crystalline polymers. It is evident to understand that compared with the regular arranged molecular in lamellae, the tie molecules (including tie chain and chain entanglements) in amorphous phase are more susceptible to relax under the influence of external forces. Therefore, the more tie molecules in the amorphous phase, the more material can be sucked into the fibrils, leading to the decrease of the stress exerted on the crack by water droplets. The essence of the crosslinking reaction is to change the molecular chain from a linear structure to a network structure. The crosslinking reaction inhibits the ordered arrangement of macromolecular chains, resulting in more local segments of macromolecular chains being excluded from the amorphous phase and increasing the density of tie molecules.

## 4. Conclusions

The fatigue mechanism for the water tree resistance of UV-photoinitiated XLPE was specifically studied using DMA analysis. Employing a UV irradiation technique, the crosslinking of polyethylene molecules was substantially ameliorated, resulting in an attenuated water tree as verified by the accelerated aging experiment. The DMA analyses demonstrated that the β-relaxation of XLPE representing activity of an amorphous phase has been appreciably enhanced with the considerable increase of crosslinking degree caused by UV irradiation. For different crosslinking degrees, the water tree resistance of XLPE was elucidated by correlating the fibril length, which was evaluated using the loss factor and loss modulus of the dynamic thermomechanical analysis. Based on the fatigue mechanism of water tree growth in a semi-crystalline polymer, it was reasonably suggested from experiment results that the crosslinking reaction could effectively delay water tree growth, which was essentially attributed to the large number of interconnecting molecular chains generated by activating the crosslinking bonds in the region of the amorphous phase under UV irradiation. The increased length of crack fibers could reduce the damaging stress from the microbeads so as to hinder water tree development.

## Figures and Tables

**Figure 1 materials-12-00746-f001:**
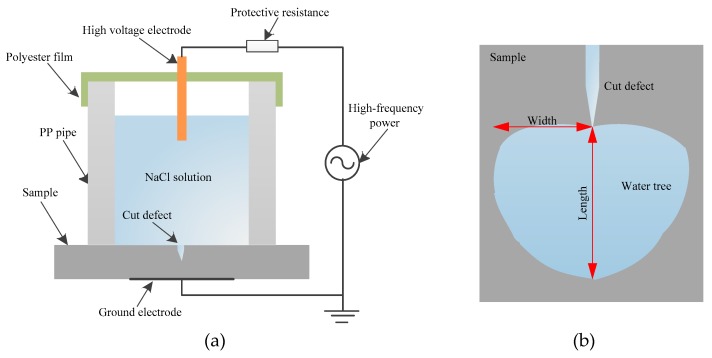
Schematic water knife electrode system: (**a**) experimental installation, and (**b**) water tree region.

**Figure 2 materials-12-00746-f002:**
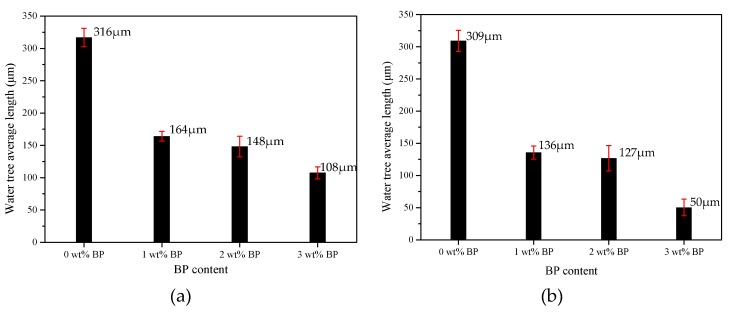
Water tree size (**a**) length and (**b**) width.

**Figure 3 materials-12-00746-f003:**
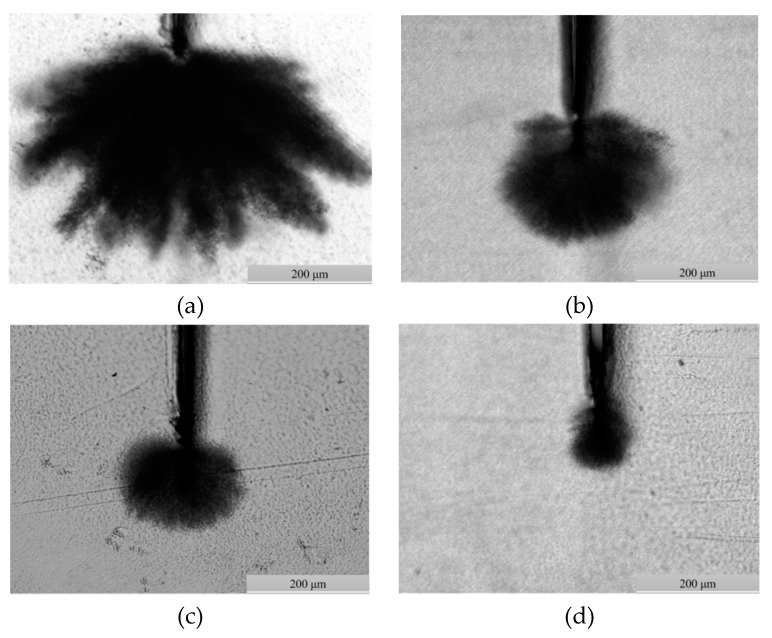
Water tree morphology of XLPE with different crosslinking degree: (**a**) XLPE-0 wt % BP, (**b**) XLPE-1 wt % BP, (**c**) XLPE-2 wt % BP and (**d**) XLPE-3 wt % BP.

**Figure 4 materials-12-00746-f004:**
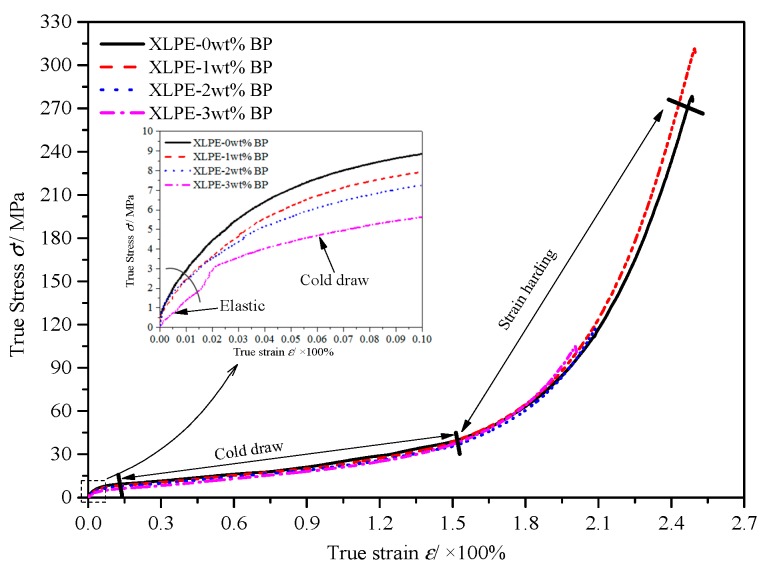
True stress–strain curves.

**Figure 5 materials-12-00746-f005:**
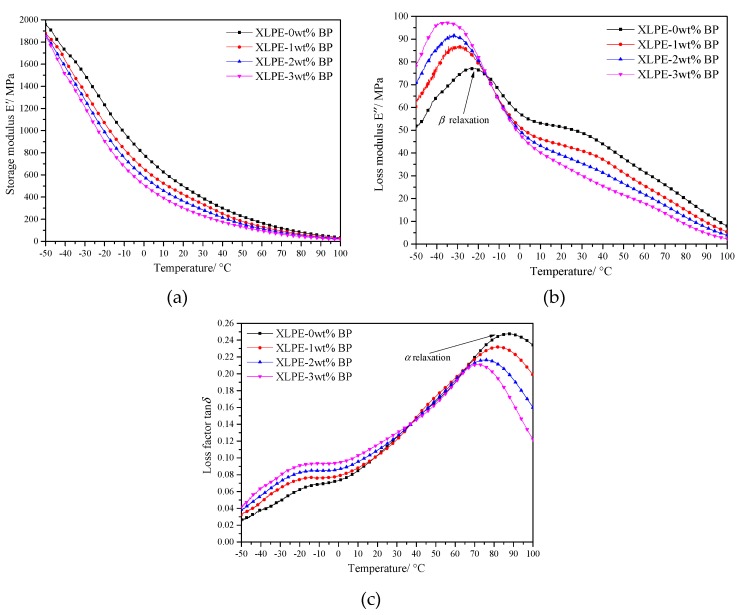
Dynamic relaxation temperature spectra: (**a**) storage modulus *E*′; (**b**) loss modulus *E*”, and (**c**) dissipation factor tanδ.

**Figure 6 materials-12-00746-f006:**
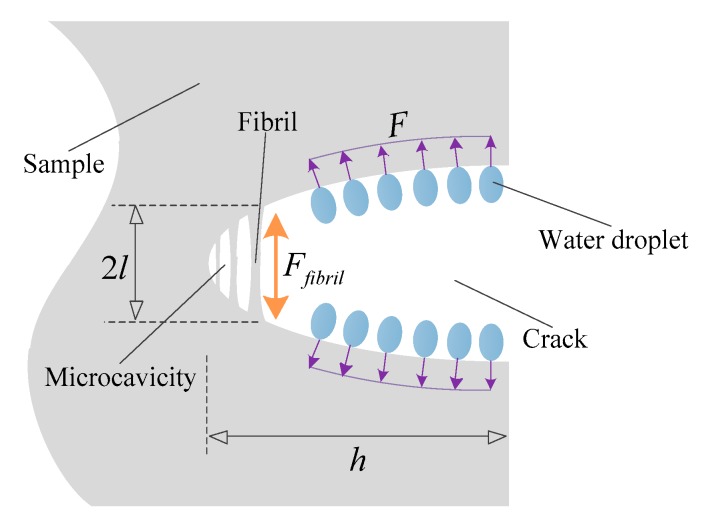
Schematic of the crack in the uniaxial stress imposed by water droplets.

**Figure 7 materials-12-00746-f007:**
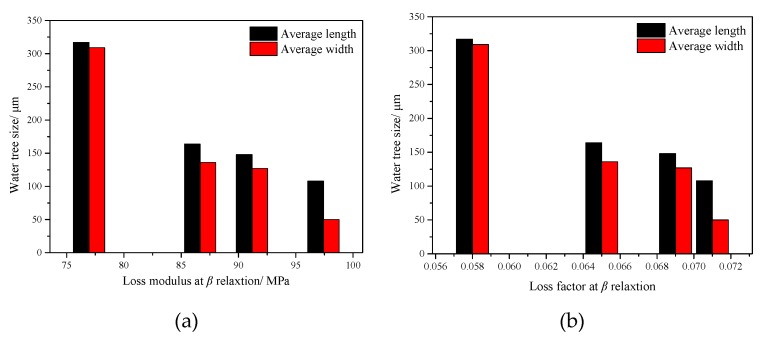
Relationship between β relaxation intensity and water tree size (**a**) loss modulus (**b**) loss factor.

**Table 1 materials-12-00746-t001:** Thermal elongation and gel content of materials.

Samples	Elongation (%)	Gel Content (%)
XLPE-0 wt % BP	Broken	0
XLPE-1 wt % BP	100	70
XLPE-2 wt % BP	40	85
XLPE-3 wt % BP	10	94

**Table 2 materials-12-00746-t002:** Characteristic parameters of elongation performance.

Samples	Elastic Modulus (MPa)	Breaking Stress (MPa)	Breaking Elongation (%)	Strain Hardening Index
XLPE-0 wt % BP	569.2	105.1	248	6.22
XLPE-1 wt % BP	459.9	117.9	249	6.29
XLPE-2 wt % BP	234.5	308.1	209	6.60
XLPE-3 wt % BP	196.1	277.7	200	6.64

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
