# Peer review of "Dynamic Thermomechanical Analysis on Water Tree Resistance of Crosslinked Polyethylene"

_materials, 2019, doi:10.3390/ma12050746_

Round 1
Reviewer 1 Report
The paper submited by Sun et al. is generally well-written, however I have some minor suggestions on the paper.
Comments:
Authors used for blending the torque rheometer, however it is not written if it was co-rotating or counter-rotating. And since it affects the diespersion of the fillers, such infrmation should be provided.
Point 2.3. Please provide what standard has been followed in the tensile test.
Authors observed the effects in the cables, however is ot possible to observe such effects in the HV cables? If so, do you plan to provide such observations for different curing agents? Did you check if the similar mechanism might be found in the screen in the cables, which contains nanofillers. Do you plan to extend this study?
Author Response
Response to Reviewer 1 Comments
Thank you for your amendments and addenda to this paper. We have made the following improvments to the manuscript in accordance with your comments.
Point 1: Authors used for blending the torque rheometer, however it is not written if it was co-rotating or counter-rotating. And since it affects the dispersion of the fillers, such infrmation should be provided.
Response 1: In order to guarantee the fillers uniform disperse in PE, the rotation mode of screws in torque rheometer is counter-rotating. The statement in 60th line (section 2.1) “Firstly, LLDPE, BP and TAIC are melted and being blended for 20min by torque rheometer with the rotational speed of 60rmp at the mixing temperature of 160℃”, has been modified as “Firstly, LLDPE, BP and TAIC are melted and blended by torque rheometer to obtain the crosslinkable materials. The rotation mode of screws in torque rheometer is counter-rotating with the rotational speed of 60rmp at the mixing temperature of 160℃ for 20 min in order to make BP and TAIC dispersed uniformly in polyethylene”.
Point 2: Please provide what standard has been followed in the tensile test.
Response 2: The standard in the tensile test is SAC Publication GB/T 1040.2-2006.The statement in 99th line (section 2.3) “The multifunctional electronic tensile testing machine-CMT6000 manufactured by Meters Industrial Systems Co. Ltd. is employed to measure mechanical properties of the samples in standard "5A" dumbbell shape of GB/T 1040.2-2006, in which the displacement rate is set to 5mm/min”, has been modified as “The multifunctional electronic tensile testing machine-CMT6000 (manufactured by MTS Industrial Systems Co., Ltd., China) is employed to measure mechanical properties of the samples according to SAC Publication GB/T 1040.2-2006. The test samples were molded into "5A" dumbbell shape with 1mm thickness and 20mm gauge length. The tensile displacement rate of samples is set to 5mm/min”.
Point 3: Authors observed the effects in the cables, however is or possible to observe such effects in the HV cables?
Response 3: Does the “effects” refers to the conclusion that crosslinking reaction can influence the anti-water-tree resistance of PE or XLPE? In general, there is a difference in crosslinking degree along the thickness direction of HV cable insulation due to the different temperature of insulation along its thickness direction during manufacture processing as shown in Figure 1 provided by a cable manufacturer. So even though we did not perform the water-tree aging test on HV cable XLPE insulation, we confirm that there is a difference in anti-water-tree resistance in XLPE insulation along its thickness direction and the analysis results may be the same as the content of this paper.
(a) (b)
(a) Temperature distribution in XLPE insulation during manufacture processing
(b) Crosslinking degree curve in XLPE insulation
Figure 1
Point 4: Do you plan to provide such observations for different curing agents?
Response 4: It is well known that curing agents would change the aggregation structure of PE or XLPE, so there is no doubt that the anti-water-tree resistance will be influenced by curing agents. We are carrying out this research at present.
Point 5: Did you check if the similar mechanism might be found in the screen in the cables, which contains nanofillers. Do you plan to extend this study?
Response 5: Water tree will not appear in the screen in the cables. The screen in the cables is semiconductive so there is no nonuniform electric field in screen, which is requirement for the initiation and growth of water tree. However, characteristic of the interface between screen and XLPE insulation may have an effect on the growth of water trees in XLPE insulation. Therefore, we will extend this study in further.
Reviewer 2 Report
The english has a few errors/unclear statements which should probably be proofed and revised before publication (e.g. chemically jointing, flaked out, the accelerate water ageing etc).
"Power cable" is usually written "power cables". e.g. "insulation materials for power cables", "medium voltage power cable" etc
Water treeing should be described (in one or two lines) early in the paper.
It is written that the "conclusions need to be confirmed by further discussion". Conclusions cannot be confirmed by discussing them.
The authors should consider moving the results in Table 1 out of 2.1 and into part 3.
The tests used to measure thermal extension and gel extraction should be described.
Fig 2b should have the average width as the y axis (instead of average length)
2.4 15um should be 15μm?
Fig 4 - the initial part of the curve is not elastic, since it is clearly curved in the inset graph.
3.4 correct "Waxwel1 force"
3.4 change "Men" to "Men et al."
Author Response
Response to Reviewer 2 Comments
Thank you for your amendments and addenda to this paper. We have made the following improvments to the manuscript in accordance with your comments.
Point 1 and 2: The English has a few errors/unclear statements which should probably be proofed and revised before publication (e.g. chemically jointing, flaked out, the accelerate water ageing etc). "Power cable" is usually written "power cables". e.g. "insulation materials for power cables", "medium voltage power cable" etc
Response 1 and 2: The errors and unclear statements have been modified. Please check the "track changes" of the revised manuscript.
Point 3: Water treeing should be described (in one or two lines) early in the paper.
Response 3: Water treeing is described early in the paper (refer to 28th line of the modified manuscript). It stated as “Water tree is a degradation phenomenon for insulation material. It consists of large numbers of voids filled with water inside. It is generally believed that the generation and development of water trees in insulation material is the result of long-term interaction by nonuniform electric field and infiltrated water”.
Point 4: It is written that the "conclusions need to be confirmed by further discussion". Conclusions cannot be confirmed by discussing them.
Response 4: The statement “these conclusions need to be confirmed by further discussion” has been modified as “these research achievements need to be further analysed”.
Point 5 and 6: The authors should consider moving the results in Table 1 out of 2.1 and into part 3. The tests used to measure thermal extension and gel extraction should be described.
Response 5 and 6: The Table 1 has been moved out from 2.1 section into 3.1 section. The “thermal extension and gel extraction” test procedure has been described in 2.2 section.
Point 7: Fig 2b should have the average width as the y axis (instead of average length)
Response 7: The y axis in Fig 2b denotes the average width. The data of average width and average length is very close so that you mistake average width as average length. The height of columns was labeled in the Fig 2 to distinguish easily.
Point 8: 2.4 15um should be 15μm?
Response 8: The unit error has been modified.
Point 9: Fig 4 - the initial part of the curve is not elastic, since it is clearly curved in the inset graph.
Response 9: The curve shown in the inset graph contain elastic and cold draw. We made a label in the inset graph.
Point 10 and 11: 3.4 correct "Waxwel1 force". 3.4 change "Men" to "Men et al."
Response 10 and 11: These words error has been modified.

Reviewer 3 Report
This paper describes dynamic thermomechanical analysis (DMA) to characterize the microstructure of polyethylene before and after cross-linking reaction and explain the intrinsic mechanism of crosslinking reaction to improve water tree resistance of polyethylene. The authors examined true stress-strain characteristics and viscoelastic properties of polyethylene cross-linked with combination of benzophenone and triallyl isocyanurate. I think the experiments were carried out carefully. This work will give useful information in the field of application of cross-linked polyethylene. The only thing I am worrying about is insufficient characterization of the cross-linked polyethylene. Probably, many readers want to know the effect of BP content on cross-linking density. I think swelling property experiment will help the enhancement of the quality of this paper.
Author Response
Response to Reviewer 3 Comments
Thank you very much for your praise to the content of this paper. We have made the following improvments to the manuscript in accordance with your comments.
Point 1: The only thing I am worrying about is insufficient characterization of the cross-linked polyethylene. Probably, many readers want to know the effect of BP content on cross-linking density.
Response 1: The crosslinking mechanism of UV irradiation crosslinking polyethylene has been added to this paper in 2.1 section. The effect of BP is absorbing UV light to form free radicals and initiating crosslinking reaction between PE molecule chains. The effect of TAIC is accelerating the rate of crosslinking reaction. So the more BP mixed in the crosslinkable materials, the higher crosslinking degree and crosslinking reaction rate will be obtained. The detailed crosslinking mechanism refers to the modified manuscript.
Point 2: I think swelling property experiment will help the enhancement of the quality of this paper.
Response 2: The swelling property experiment is helpful to understanding the aggregation structure of PE and XLPE. In this paper, we adopt gel extraction method to evaluate the crosslinking degree of samples, but the analysis on swelling property is slightly insufficient to some extent. We will intensive study the swelling property of PE and XLPE in the future to improve the research. Thank you for your comments.

This manuscript is a resubmission of an earlier submission. The following is a list of the peer review reports and author responses from that submission.